# Protective Effects of Epigallocatechin-3-gallate (EGCG) against the Jellyfish *Nemopilema nomurai* Envenoming

**DOI:** 10.3390/toxins15040283

**Published:** 2023-04-14

**Authors:** Jie Li, Qianqian Wang, Shuaijun Zou, Juxingsi Song, Peipei Zhang, Fan Wang, Yichao Huang, Qian He, Liming Zhang

**Affiliations:** 1Department of Marine Biomedicine and Polar Medicine, Naval Special Medical Center, Naval Medical University, Shanghai 200433, China; lijie1992@smmu.edu.cn (J.L.);; 2Department of Marine Biological Injury and Dermatology, Naval Special Medical Center, Naval Medical University, Shanghai 200052, China; 3The Third Affiliated Hospital, Naval Medical University, Shanghai 200433, China

**Keywords:** epigallocatechin-3-gallate (EGCG), jellyfish venom, hemolytic toxicity, proteolytic activity, cardiomyocyte toxicity

## Abstract

Jellyfish stings are the most common marine animal injuries worldwide, with approximately 150 million envenomation cases annually, and the victims may suffer from severe pain, itching, swelling, inflammation, arrhythmias, cardiac failure, or even death. Consequently, identification of effective first aid reagents for jellyfish envenoming is urgently needed. Here, we found that the polyphenol epigallocatechin-3-gallate (EGCG) markedly antagonized the hemolytic toxicity, proteolytic activity, and cardiomyocyte toxicity of the jellyfish *Nemopilema nomurai* venom in vitro and could prevent and treat systemic envenoming caused by *N. nomurai* venom in vivo. Moreover, EGCG is a natural plant active ingredient and widely used as a food additive without toxic side effects. Hence, we suppose that EGCG might be an effective antagonist against systemic envenoming induced by jellyfish venom.

## 1. Introduction

Jellyfish are gelatinous invertebrate zooplankton in aquatic environments, with large numbers and a wide distribution around the globe. Jellyfish are composed of three parts: bell, tentacle, and oral arm. The tentacles and oral arms are equipped with a special type of stinging cells, the nematocytes, which are used for capturing prey or defense [1]. The nematocyte contains mainly an organoid, the nematocyst, which consists of a polymeric capsule wall with an inverted tubule and fluid matrix incorporating various toxins [2]. Upon exposure to a chemical or mechanical stimulus, the cnidocil is solicited, and the tubule is pushed out by the osmotic pressure created inside the cell and eventually penetrates the object, which stimulates the cnidocil [3].

In recent decades, jellyfish blooms have frequently occurred worldwide, which may be associated with global warming, overfishing, eutrophication, industrialization, and biodiversity changes, resulting in continuous stresses on the marine environment, fishery production, tourism industry, and medical health [4,5]. Moreover, jellyfish stings are the most common marine animal injuries, with approximately 150 million envenomation cases annually, and the victims may suffer from severe pain, itching, swelling, inflammation, arrhythmias, cardiac failure, or even death [6,7]. *Nemopilema nomurai*, one of the largest jellyfish in the world, is widely distributed in the east Asian marginal seas, such as the Bohai Sea, Yellow Sea, and East China Sea, causing many jellyfish sting cases every year [8]. According to clinical presentations of the patients with jellyfish stings, the toxicity of *N. nomurai* is empirically higher than that of another venomous scyphozoan in China, *Cyanea nozakii* [6]. In most cases, *N. nomurai* stings induce mild to moderate topical symptoms, including varying degrees of pain and itching with lined redness and oedema. However, severe limb swelling, systemic disorder, and death have also been reported in some severe cases of *N. nomurai* stings [9,10,11]. The composition of jellyfish venom is not known in detail, but it appears to contain a variety of proteinaceous (peptides, proteins, and enzymes) and nonproteinaceous compounds (biogenic amines, purines, and quaternary ammonium compounds) [3]. In recent years, research on antidotes against jellyfish stings has attracted increasing attention, and some progress has been made, but there is no broad consensus on the treatment measures. The way by which jellyfish venoms exert their toxic effects is yet to be fully clarified, which may help to develop specific treatments for jellyfish stings [1].

The injuries provoked by *N. nomurai* envenomation are characterized by hemolytic toxicity [12], cardiotoxicity [13,14], neurotoxicity [15], myotoxicity [16], and cytotoxicity [13]. Moreover, *N. nomurai* venom (NnV) possesses profound metalloproteinase and phospholipase-like activities, which is obviously different from other jellyfish venoms in biochemical characteristics [17,18]. Hemolysis, circulatory failure, and multi-organ damage caused by hemolytic toxicity, cardiotoxicity, and proteolytic activity are the leading causes of death from jellyfish stings [3,7]. To neutralize the toxic effects of jellyfish venom, we have been actively screening for potential antagonists and found that the polyphenol epigallocatechin-3-gallate (EGCG) can completely antagonize the hemolytic toxicity of NnV in vitro. EGCG is a major component of polyphenols from green tea, with a series of biological activities, such as antibacterial, antiviral, antioxidant, antithrombotic, anti-inflammatory, and antitumor activities [19,20]. Several studies have demonstrated the neutralization of polyphenol compounds or polyphenol-containing plant extracts towards the toxic effects caused by snake venoms [21,22]. Furthermore, EGCG significantly inhibited the proteolytic activity of jellyfish venom in a concentration-dependent manner, and topical treatment with EGCG considerably ameliorated the dermonecrotic lesions caused by jellyfish venoms [23]. Since EGCG antagonizes multiple activities of various biotoxins, whether it can be used to treat jellyfish stings arouses our interest. In the present work, we investigated the preventive and therapeutic effects of EGCG against NnV in vitro and in vivo.

## 2. Results

### 2.1. Effect of EGCG on the Proteolytic Activity of NnV

The proteolytic activity of NnV was investigated by gelatin zymography (Figure 1). NnV showed a strong gelatinolytic activity in a dose-dependent manner. Furthermore, the enzymatic hydrolysates of gelatin consisted of a multitude of bands, suggesting that the proteolytic activity in NnV might come from multienzyme complexes (Figure 1A). After treatment with EGCG, the gelatinolytic activities of NnV and collagenase were suppressed in a concentration-dependent manner (Figure 1), suggesting that EGCG could inhibit the proteolytic activity of NnV.

### 2.2. Effect of EGCG against the Hemolytic Toxicity of NnV

As shown in Figure 2A, NnV caused concentration-dependent hemolysis of the mice erythrocytes. When the concentration of NnV reached 51.2 μg/mL, the mice erythrocyte suspension was completely hemolyzed. With increasing EGCG concentration, the hemolytic activity of the mixture of NnV and EGCG decreased rapidly, indicating that EGCG had an evident inhibitory effect on the hemolytic toxicity of NnV (Figure 2B).

### 2.3. Effect of EGCG against the Cardiomyocyte Toxicity of NnV

Since the cardiotoxicity of jellyfish venom was shown to be the major factor of acute death caused by jellyfish stings [24], we further investigated whether EGCG could reduce NnV-associated cardiomyocyte toxicity in H9C2 cells (Figure 3). The results showed that NnV exposure suppressed cardiomyocyte viability with an LC_50_ value of 260 ng/mL (Figure 3A). In the preventive intervention trial, EGCG was added 30 min before incubation with NnV, and the NnV-induced cytotoxicity in H9C2 cells was reduced in a concentration-dependent manner. When the concentration of EGCG reached 20 μM, the cytotoxicity of NnV was almost completely suppressed. Similarly, the therapeutic intervention with EGCG (30 min after incubation with NnV) also had a protective effect against NnV, although the viabilities of H9C2 cells were slightly lower than those in the preventive intervention trial (Figure 3B).

### 2.4. Effect of EGCG on the Survival Rate of Mice Envenomated by NnV

The in vivo toxicity of NnV (0.29~0.6 mg/kg) was evaluated following intravenous injection in mice. As shown in Figure 4A, the envenoming symptoms and time of death of the mice were closely related to the doses of NnV. When lower doses of NnV (≤0.42 mg/kg) were injected, the mice showed a mild stress reaction at the initial stage and gradually became sluggish and unresponsive and curled up after 10 min. Some mice died in succession within 3 days, and all the surviving mice remained alive on Day 7. When moderate doses of NnV (0.42~0.6 mg/kg) were injected, shortness of breath, increased heartbeat, fatigue, malaise, and crouching appeared in the early stage, followed by gradual appearance of convulsions and insensitivity to pain stimulation, and all the mice died within 1 day. After high doses of NnV (≥0.6 mg/kg) were injected, the mice quickly developed deep and fast breathing, cyanosis, limb twitching, and angular arch back-stretching and died quickly, within 1 h. The calculated LD_50_ of NnV was 0.401 mg/kg within one week. In the preventive intervention trial, the survival time of the EGCG intervention groups was significantly prolonged compared with that of the NnV group, and the mortality was decreased dose-dependently by EGCG. When the dose of EGCG reached 20 mg/kg, all the mice survived (Figure 4B). In the therapeutic intervention trial, early administration of EGCG exhibited a noticeable antagonistic effect, but the time window for effective therapy was restricted to a narrow range of approximately 10 min (Figure 4C).

### 2.5. Effect of EGCG on the Changes in Serum Biochemical Indices Induced by NnV

Blood samples were collected to evaluate the damage to organ function, and the results showed that NnV produced significant toxic effects according to the serum biochemical indices (Figure 5). In the preventive intervention trial, the liver functional indices ALT and AST, the heart functional indices LDH and CK, and the kidney functional indices Cr and BUN all increased significantly 1 h after NnV administration, and in particular, the liver functional indices ALT and AST increased by approximately 100 times. In addition, the concentration of blood K^+^ increased, which might be a result of hemolysis caused by NnV. As expected, EGCG significantly improved the indices of heart function, liver function, and kidney function compared with those of the NnV-envenoming group. One hour or 72 h later, there were no significant differences between the blood biochemical indices of the NnV and EGCG mixture groups and the PBS group.

In the therapeutic intervention trial, as expected, all the values of the liver, heart, and kidney functional indices increased after NnV (0.32 mg/kg) administration. Although EGCG was administered 30 min after injection of NnV, the indices of heart function, liver function, and kidney function were all improved. It is worth noting that there were significant differences between the blood biochemical indices of the EGCG intervention group and the PBS group (Figure 6).

### 2.6. Histopathological Results

Light microscopy showed that the organs (liver, heart, lung, and kidneys) from the PBS control group had a normal morphology (Figure 7A,E,I,M), but different degrees of morphological changes emerged in the NnV (0.5 mg/kg) envenoming group. One hour after NnV injection, evident lesions of liver tissue (Figure 7B), including swelling, a completely destroyed lobular structure, extensive hemorrhage, and coagulation necrosis, were present. In contrast, cardiac lesions were limited (Figure 7F), primarily involving eosinophilia of myocardial cells, pyknosis of the nucleus, and congestion of interstitial capillaries. In the lungs, only a few neutrophils infiltrated (Figure 7J), and in the kidneys, there was no significant change (Figure 7N). In the NnV and EGCG mixture treatment group, these pathological changes appearing in the NnV envenoming group were all significantly improved at the early stage (1 h after treatment), and even 3 days later, there was still no obvious pathological change in the mixture treatment group.

In the therapeutic intervention trial, liver tissue lesions (Figure 8A), including hepatic necrosis, severe congestion of the liver, and hemorrhage, were also present at the early stage (1 h after NnV injection) in the NnV (0.32 mg/kg) envenoming group. Twenty-four hours later (Figure 8C), the pathological changes in liver tissue were more serious, with extensive bridging necrosis between portal areas, severe hepatic coagulation necrosis, massive hemorrhage, and neutrophil infiltration. However, there was no significant change in the heart, lungs, and kidneys. As expected, 2 h and 24 h after EGCG intervention (Figure 8B,D), the pathological changes appearing in the NnV envenoming group were all significantly improved, except for mild hepatocellular necrosis and scattered neutrophil infiltration.

## 3. Discussion

Currently, there is an increase in jellyfish stings in coastal areas worldwide. Approximately 100 people die from jellyfish stings every year in coastal countries, such as Australia, Thailand, the Philippines, and Malaysia [25]. Although only a few species of jellyfish are poisonous to humans, the hazard from jellyfish should not be ignored because the stings by some species induce serious toxic symptoms [3]. Jellyfish venoms have complex components and extensive biological activities, among which hemolytic toxicity, enzymatic activity, and cardiotoxicity are the main factors that lead to severe symptoms and even deaths from jellyfish envenomation. Hemolytic toxin, one of the few toxins that has been purified from jellyfish venom, is considered to be the major toxic factor of jellyfish venom [13], and pore formation and peroxidation are presumed to be important mechanisms of hemolysis of jellyfish venom [26].

The development and application of omics technology promotes our understanding of jellyfish venom, and it was found that proteases comprise a high proportion of jellyfish venom. Leung et al. found that there were seven types of matrix metalloproteinases (MMPs) and nine types of MMPs in *Rhopilema esculentum* and *Sanderia malayensis*, respectively [27]. Choudhary et al. found that metalloproteinases accounted for 21% of the toxin proteins in *N. nomurai* venom [28]. Yang et al. found 120 toxin-related proteins, including 31 metalloproteinases, in the toxin transcription group and 62 toxin-related proteins, including 16 metalloproteinases, in the protein group of jellyfish *Phacellophora camtschatica* [29]. The role of proteases in dermal toxicity [30], myotoxicity [16], and cytotoxicity [31] has received increasing attention. In recent years, drugs that antagonize the MMP activity of jellyfish venom, such as tetracycline, have been proven to be effective in the treatment of cutaneous damage induced by jellyfish stings [30].

Therefore, we carried out drug screening for potential antagonists against the hemolytic toxicity and proteolytic activity of jellyfish venom. The results showed that EGCG, a type of green tea polyphenol that can inhibit the toxic effects of various toxins, such as snake toxin, botulinum toxin, tetanus toxin, and Shiga toxin [32,33,34,35], significantly inhibited the proteolytic activity of NnV. Furthermore, we tested the intervention effect of EGCG on hemolysis and cardiomyocyte toxicity and were excited to find that EGCG effectively antagonized the hemolytic toxicity of NnV and protected H9C2 cells from the cytotoxicity of NnV. Therefore, the administration of EGCG after jellyfish envenomation may be effective.

We further investigated the antagonistic effect of EGCG in vivo. In the preventive intervention trial, EGCG completely prevented the death of mice caused by NnV and significantly improved the indices of heart function, liver function, and kidney function and histopathological lesions of the liver and heart compared with those of the NnV envenoming group. In the therapeutic intervention trial, EGCG also improved the blood biochemical indices and pathological damage.

## 4. Conclusions

In summary, EGCG can antagonize the hemolytic toxicity, proteolytic activity, and cardiomyocyte toxicity of NnV in vitro and in vivo. Since EGCG is a natural plant active ingredient and widely used as a food additive without toxic side effects [23], our results imply that EGCG may be useful for the treatment of systemic envenoming induced by jellyfish stings.

However, in the acute lethal model where mice were intravenously injected with 0.5 mg/kg NnV, we found that the therapeutic effect of EGCG decreased rapidly with the delay of EGCG administration, which may be due to the following two reasons. First, large doses of NnV can cause irreversible organic damage at an early stage, which weakens the therapeutic effect of EGCG. Second, EGCG is easily inactivated by aggregation with complex components in blood, such as erythrocyte membrane and albumin, and easily transformed by metabolic enzymes. The above characteristics and self-oxidation tendency of EGCG may cause a decrease in the bioavailability of EGCG in vivo [36]. Nevertheless, in view of the prominent antagonistic effect of EGCG against jellyfish venom in vitro and in vivo, it may be worthwhile to construct an efficient drug delivery system to improve the bioavailability of EGCG and give full play to its therapeutic effect on systemic envenoming by jellyfish venom.

## 5. Materials and Methods

### 5.1. Reagents and Jellyfish Collection

EGCG (purity 95%) was purchased from Shanghai Macklin Biochemical Co., Ltd. Percoll (1.129 g/mL) was purchased from GE Healthcare Life Sciences. Phosphate buffered saline (PBS) was purchased from Shanghai Beyotime Biotechnology Co., Ltd. (Shanghai, China). Heparin sodium (150 u/mg) was purchased from Shanghai Yuanye Biotechnology Co., Ltd. Cell Counting Kit-8 was purchased from Dojindo China Co., Ltd. (Shanghai, China). Fetal bovine serum (FBS) and pen strep were obtained from Thermo Fisher Scientific Co., Ltd. (Waltham, MA, USA).

Specimens of the jellyfish *Nemopilema nomurai* were collected from Laoshan Bay in Qingdao, China, in August 2020. The jellyfish tentacles were excised immediately with scissors from living specimens. The isolated tentacles were placed in plastic bags with dry ice and immediately transported back to the laboratory, where the samples were stored in a −80 ℃ freezer until use.

### 5.2. Nematocyst Isolation

Nematocysts were isolated from *N. nomurai* tentacles according to the method described by Wang et al. [7] with a slight modification. Briefly, the frozen tentacles were thawed at 4 °C in artificial seawater (NaCl 28 g, MgCl_2_·6H_2_O 5 g, KCl 0.8 g, and CaCl_2_ 1.033 g, added distilled water to 1000 mL) at a quality and volume ratio of 1:1 for autolysis. After 4 d, the mixtures were filtered through a 200-mesh sieve to remove large tissue debris. The filtrate was then centrifuged at 500× *g* for 3 min at 4 °C, and sediments were washed three times with artificial seawater. The 50% and 100% Percoll solutions prepared with artificial seawater were placed at 3 mL per concentration from high to low in a 15 mL centrifuge tube, and then, the washed sediments were loaded at 3 mL. The centrifuge tube was centrifuged horizontally at 1000× *g* for 15 min at 4 °C, and then, the white sediments at the bottom were collected and washed three times with artificial seawater to obtain nematocysts containing venom. The nematocysts were used immediately for venom extraction or, alternatively, frozen at −80 °C until use.

### 5.3. Venom Extraction

Jellyfish venom was extracted from nematocysts using the method by Li et al. [37] with a slight modification. Briefly, the nematocysts were suspended in deionized water and then ultrasonicated using a Misonix Sonicator (S-4000-010, Qsonica LLC, The Meadows, FL, USA) at 400 W until all the nematocysts were released or broken under a microscope. After centrifugation at 12,000× *g* for 15 min at 4 °C, the supernatant, which comprised the nematocyst venom of *N. nomurai*, was then dialyzed against phosphate buffer saline (PBS) (0.01 mol/L, pH 7.4) for 8 h before use. The nematocyst venom obtained from the jellyfish *N. nomurai* was named NnV, and the protein concentration of NnV was determined by the Bradford method.

### 5.4. Animal Maintenance

Male ICR mice (20 ± 2 g) were provided by the laboratory animal center of Naval Medical University, Shanghai. The animals were kept in an animal care facility under controlled temperatures and normal day and night cycles, and all animals were acclimatized for one week in laboratory conditions before carrying out the experimental study. The investigation was carried out in conformity with the requirements of the Ethics Committee of the Naval Medical University.

### 5.5. Proteolytic Activity Assay

Gelatin was used as a substrate for the proteolytic zymography assay as described by Lee et al. [31], and the gelatin enzyme profiling kit was purchased from Real-Times (Beijing, China). Gelatin (2 mg/mL) was dissolved in 20 mM sodium phosphate buffer (pH 7.4) and copolymerized with 10% polyacrylamide to prepare zymography gels. Venom extracts (2 and 4 μg) to be analyzed were prepared in nonreducing sample buffer and then run on gels at 150 V/gel for 1 h at 4 °C. After electrophoresis, the gel was washed for 30 min twice with renaturing buffer and incubated for an additional 18 h at 37 °C for enzymatic reaction in zymography reaction buffer. The gel was then stained with FastBlue staining solution. Clear zones in the gel indicate regions of proteolytic activity. For investigation of the inhibitory effect of EGCG on the proteolytic activity of NnV, the different concentrations of EGCG (0, 1, 2, and 4 mM) were added to all the wash and incubation buffers, and the gel was stained as described above.

### 5.6. Hemolysis Test

In brief, freshly collected blood samples from mice were immediately mixed with anticoagulant (1% heparin in PBS, pH 7.4) to prevent blood coagulation. For a pure suspension of erythrocytes, 1 mL of whole blood was made up to 20 mL in phosphate buffered saline (PBS, pH 7.4) and centrifuged at 1000× *g* for 10 min at 4 °C. The supernatant and buffy coats were then discarded by gentle aspiration, and the erythrocyte pellet was washed twice and suspended in the same buffer to a final concentration of 0.5% (*v*/*v*) [38]. Various concentrations of NnV (0.05, 0.2, 0.8, 3.2, 12.8, 51.2, and 204.8 μg/mL) were added to the erythrocyte suspension (100 μL, 0. 5% in 0.01 M phosphate buffer). The total volume of the test system was 200 μL. The samples were incubated at 37 °C for 30 min in a water bath. The intact erythrocytes and erythrocyte ghosts were removed by centrifugation at 1500× *g* for 5 min at 4 °C. A 150 mL portion of the supernatant fluid was transferred to a 96-well microplate, and its optical absorbance (OD) was measured at 540 nm using a spectrophotometric microplate reader (Thermo Fisher). The concentration of the hemoglobin released from lysed erythrocytes was taken as the index of NnV-induced hemolysis. The negative (PBS) and positive (30 mg/mL saponin) controls were taken as 0% and 100% hemolysis, respectively. The hemolytic activity of NnV was expressed as % absorbance compared with that of the positive control group.

According to the results of the NnV hemolysis test, the appropriate concentration of NnV was selected and then premixed with various concentrations of EGCG (0, 31.25, 62.5, 125, 250, 500, 1000, and 2000 μM) for 1 min. The hemolytic activity of the mixture was tested by the above method.

### 5.7. Cell Culture and Cytotoxicity Test

Cell viability was measured by the Cell Counting Kit-8 assay as previously described [7]. Briefly, rat cardiomyocyte (H9C2) cells purchased from Shanghai Anwei Biotechnology Co., Ltd (Shanghai, China) were grown in DMEM with 10% FBS and 100 mg/mL penicillin–streptomycin at 37 °C in a 5% CO_2_ humidified incubator. H9C2 cells were incubated at a density of 7 × 10^4^ cells/well in 96-well plates for 24 h in 100 μL of complete DMEM. After an adaptation period, nonadherent cells were removed by gentle washing with fresh culture medium, and the remaining cells were treated with NnV at the indicated concentrations (0~500 ng/mL). After incubation for another 4 h, cytotoxicity was evaluated by measuring mitochondrial dehydrogenase activity using Cell Counting Kit-8 assays. CCK-8 (10 μL) was added to each well, and the plates were incubated for 2 h at 37 °C. Then, the amount of formazan salt was determined by measuring the optical density (OD) at 450 nm using a spectrophotometric microplate reader (Thermo Fisher, Waltham, MA, USA).

For investigation of the protective effect of EGCG against NnV cytotoxicity, the cells were treated with different doses of EGCG (0~20 μM) for 30 min before or after the addition of the LC_50_ dose of NnV, and then, 4 h after NnV addition, cell viability was tested by the above method.

### 5.8. Survival Analysis

Male ICR mice (20 ± 2 g) were randomly divided into five groups (*n* = 10) and given different doses of NnV, with a ratio of adjacent concentration of 1.2, through the tail vein and then observed for one week. The envenoming symptoms and the number of dead mice were recorded, and the median lethal dose (LD_50_) was calculated. In the preventive experiment, a 1.25× LD_50_ dose of NnV was premixed with different doses of EGCG (5~20 mg/kg), and then, the mice were injected with the mixtures via the tail vein and observed for one week. In the therapeutic experiment, after a 1.25× LD_50_ dose of NnV was injected into mice via the tail vein, the mice were injected with EGCG at different intervals (5, 10, and 20 min) and observed for one week.

### 5.9. Blood Biochemical Analysis

In a preventive intervention trial, male ICR mice (20 ± 2 g) were injected with a 1.25× LD_50_ dose of NnV alone or in combination with EGCG (the negative control group was injected with an equal volume of PBS). One hour or 72 h later, blood samples were collected from the retro-orbital venous plexus. The collected blood samples were stored for 2 h at 4 °C and then centrifuged at 2000× *g* for 15 min. The clear supernatants were collected and used to determine blood biochemical indicators, including the heart indices LDH and CK, liver indices ALT and AST, kidney indices Cr and BUN, and electrolyte indices Ca^2+^, Na^+^, K^+^, and Cl^−^.

In a therapeutic intervention trial, a 0.8× LD_50_ dose of NnV was injected via the tail vein, and 30 min later, the mice were injected with PBS or EGCG. After injection with NnV for 2 h and 24 h, blood samples were collected, and the blood biochemical indicators were analyzed by the above method.

### 5.10. Histological Examination

In the preventive intervention trial, male ICR mice (20 ± 2 g) were injected with a 1.25× LD_50_ dose of NnV alone or in combination with EGCG. One hour later, the animals were sacrificed by cervical dislocation. The heart, liver, lung, and kidney tissues were dissected and immediately placed in PBS (10 mM, pH 7.4) containing 4% paraformaldehyde for 24 h. Then, the tissues were dehydrated and embedded in paraffin. The tissue blocks were sectioned and stained with hematoxylin and eosin (H&E) for the observation of histological changes.

In the therapeutic intervention trial, a 0.8× LD_50_ dose of NnV was injected via the tail vein, and 30 min later, the mice were injected with PBS or EGCG. After injection with NnV for 2 h and 24 h, the tissues were collected, and histological changes were examined by the above method.

### 5.11. Statistical Analysis

In the experiments, all values in the figures and text are expressed as the mean ± SD. The significance of differences between the means of various experimental groups was analyzed by Analysis of Variance (ANOVA), followed by Dunnett’s *t*-test using SPSS Statistics 22.0 software (IBM, Armonk, New York, NY, USA). 

## Figures and Tables

**Figure 1 toxins-15-00283-f001:**
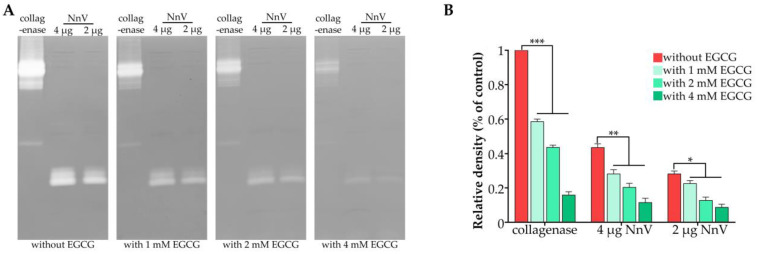
Effects of EGCG on the proteolytic activity of NnV. Collagenase and NnV (2 and 4 μg) were loaded and run on SDS–PAGE, followed by gelatin zymography in reaction buffers. (**A**) The zymograms of proteolytic enzymes in NnV under different reaction buffers, which contained increasing concentrations of EGCG (0, 1, 2, and 4 mM). (**B**) The proteolytic activity data were quantified using densitometric analysis with collagenase without EGCG as a control by ImageJ software. Data are shown as the mean ± SD from three independent experiments. * *p* < 0.05, ** *p* < 0.01, *** *p* < 0.001.

**Figure 2 toxins-15-00283-f002:**
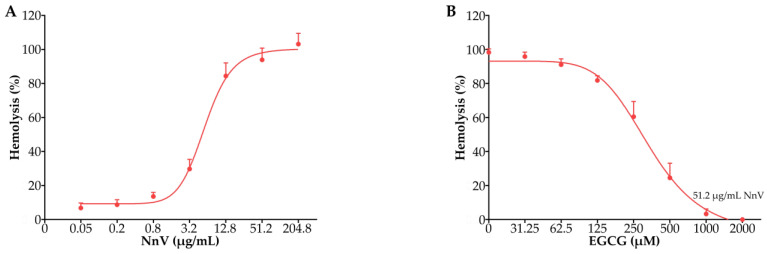
Effect of EGCG against the hemolytic toxicity of NnV in the mice erythrocytes. (**A**) Hemolytic toxicity of NnV at different concentrations (0.05, 0.2, 0.8, 3.2, 12.8, 51.2, and 204.8 μg/mL). (**B**) Inhibitory effect of EGCG (0, 31.25, 62.5, 125, 250, 500, 1000, and 2000 μM) on the hemolytic toxicity of NnV (51.2 μg/mL). Data are shown as the mean ± SD from three independent experiments.

**Figure 3 toxins-15-00283-f003:**
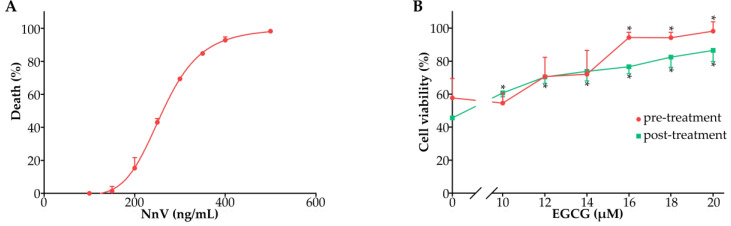
Effect of EGCG against the cardiomyocyte toxicity of NnV. (**A**) H2C9 cells were treated with NnV at different concentrations (0~500 ng/mL). (**B**) H2C9 cells were treated with different doses of EGCG (0~20 μM) 30 min before or after NnV exposure (260 ng/mL). Data are shown as the mean ± SD from three independent experiments. * *p* < 0.05 compared with the NnV alone group.

**Figure 4 toxins-15-00283-f004:**
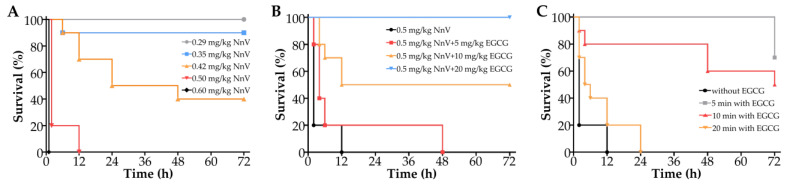
Effect of EGCG on the survival rate of mice envenomated by NnV. (**A**) The mice were intravenously injected with various doses of NnV (0.29~0.6 mg/kg) and observed for one week. (**B**) The mice were injected with NnV (0.5 mg/kg) or NnV mixed with various doses of EGCG (5, 10, and 20 mg/kg) by tail vein and observed for one week. (**C**) The mice were injected with EGCG (20 mg/kg) at different time points (5, 10, and 20 min) or without EGCG after the injection of NnV (0.5 mg/kg) and observed for one week.

**Figure 5 toxins-15-00283-f005:**
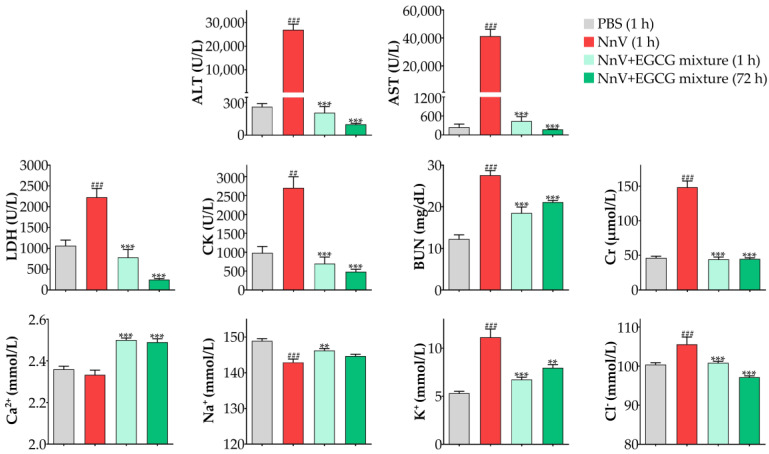
Preventive effect of EGCG (20 mg/kg) on the changes in serum biochemical indices induced by NnV (0.5 mg/kg). ^##^
*p* < 0.01, ^###^
*p* < 0.001 compared with the negative control (PBS) group. ** *p* < 0.01, *** *p* < 0.001 compared with the NnV alone group.

**Figure 6 toxins-15-00283-f006:**
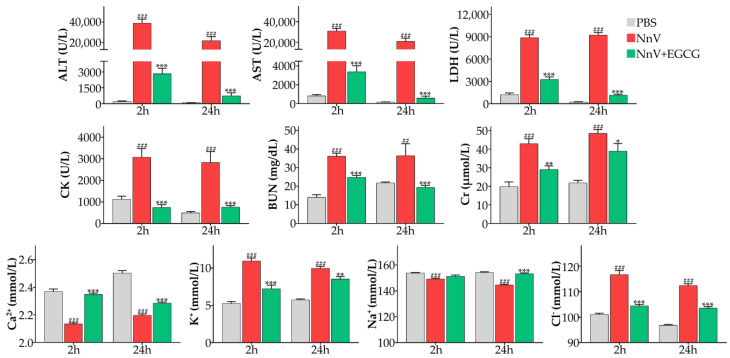
Therapeutic effect of EGCG (20 mg/kg) on the changes in serum biochemical indices induced by NnV (0.32 mg/kg). ^##^
*p* < 0.01, ^###^
*p* < 0.001 compared with the negative control (PBS) group. * *p* < 0.05, ** *p* < 0.01, *** *p* < 0.001 compared with the NnV alone group.

**Figure 7 toxins-15-00283-f007:**
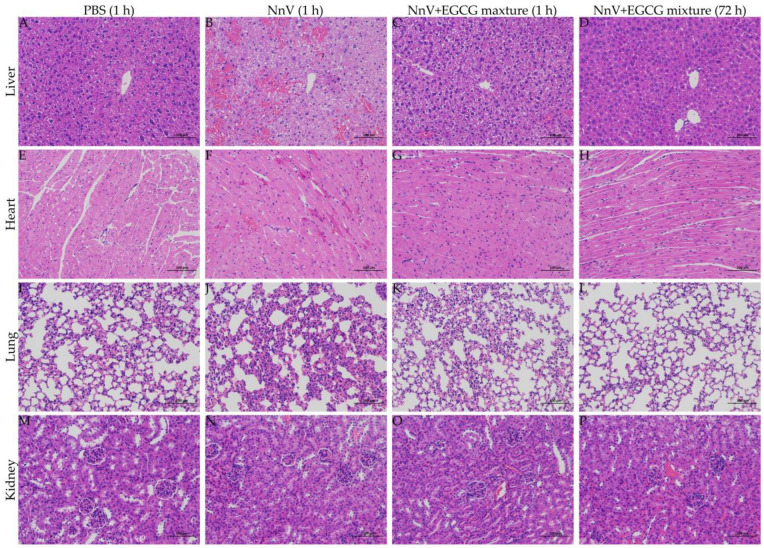
Effect of EGCG (20 mg/kg) against NnV-induced (0.5 mg/kg) organ damage in the preventive intervention trial (the scale bars in the figures represent 100 μm).

**Figure 8 toxins-15-00283-f008:**
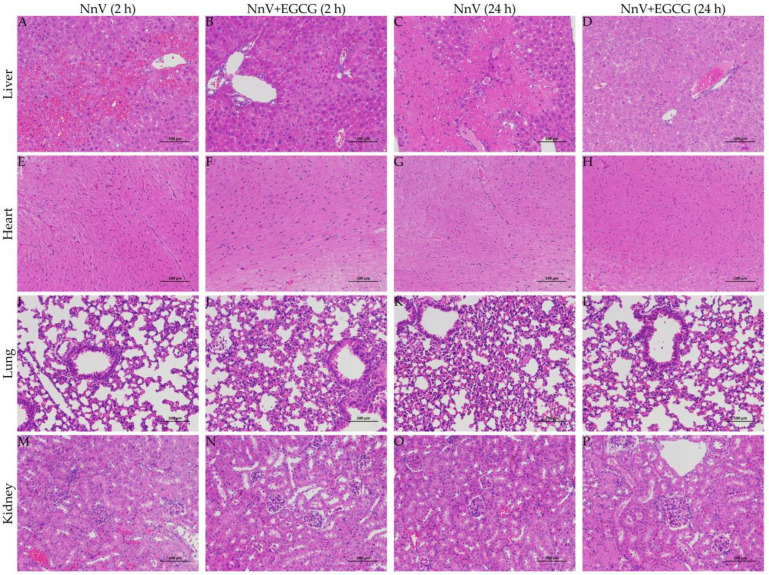
Effect of EGCG (20 mg/kg) on NnV-induced (0.32 mg/kg) organ damage in the therapeutic intervention trial (the scale bars in the figures represent 100 μm).

## Data Availability

The data presented in this study are available on request from the corresponding author.

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
