# Peer review of "Protective Effects of Epigallocatechin-3-gallate (EGCG) against the Jellyfish Nemopilema nomurai Envenoming"

_toxins, 2023, doi:10.3390/toxins15040283_

Round 1
Reviewer 1 Report
The last paragraph in the introduction should give the reader more insight into your study. Based on previous studies, what are your hypotheses/expectations for each of the things you are examining?
The qualitative nature of the histological analysis. The histological analysis might have been more valuable to the reader if some index of severity were indicated. Currently, the scale is simply “positive” or “negative”. Some measure of severity would have helped.
Discussion: The authors need to improve the discussion section focusing on the importance and relevance of this study.
Reviewer 2 Report
This manuscript, describing the protective effects of EGCG against the jellyfish Nemopilema nomurai venom, clearly and concisely presents accurate and completed data. It might be published in Toxins journal after some minor revisions.
I strongly recommend to include the more information about "jellyfish venom (NnV) " (P 2 L60) in the introduction section; since there is no data about N. nomurai venom except for P 7 L 119-200 in the manuscript. In particular the following details:
the venom source - Nemopilema nomurai, why this organism was selected as venom source?
what is the clinical features of poisoning, how typical is this effect for jellyfish in general, are there any characteristic features of the symptoms of NnV poisoning?
what is known about the composition of the poison, whether it differs in composition from the poisons of other species of jellyfish?
If there is no data in should be mentioned. The information could be given in brief in the introduction section or at the very least in the discussion section.
It is necessary to improve the quality of all figures, some graphs even lack axes.
I recommend to add to the fig 1a caption something like "Zymograms of proteolytic enzymes in the NnV venom", also add "without EGCG" instead of " EGCG-" under the first zymogram on the fig 1a and replace " EGCG-" to the "EGCG" under the three following zymograms.
P2 L78 To the sentence "As shown in Fig. 2A, NnV caused concentration-dependent hemolysis" add please "of the mice erythrocytes".
It is not clear what the authors mean in P2 L79 "the hemolysis rate exceeded 100%". It should be rephrased or explained.
Please consider adding a graph for the NnV alone group to Fig. 3B.
As mentioned in P8 L275 " protein concentration of NnV was determined by the Bradford method ". However in the introduction it was rightly noted that " jellyfish venom appears to contain a variety of proteinaceous ... and nonproteinaceous compounds (purines, quaternary ammonium compounds, biogenic amines and betaines)". Doesn't this mean that the concentration of non-protein compounds, which could be responsible for some of the observed effects, was not taken into account. It would be interesting to measure what is the content of protein components, according to Bradford, in a lyophilized sample of venom weighing 1 mg, for example.
Please explain why the NnV doses of 1.25× LD50 and 0.8× LD50 were chosen for the changes in serum biochemical indices and histological examination.
Please add the EGCG concentration in the section 4.5. Proteolytic activity assay (despite the fact that this described in the 2.1 section).
Reviewer 3 Report
The article titled " Protective effects of green tea epigallocatechin-3-gallate (EGCG) against jellyfish venom poisoning" reports interesting data about a potential alternative against the poisoning caused by the venom of the jellyfish Nemopilema nomurai.
The article is well structured, but I suggest some changes before publication:
1. Page1, line 1-3: The title of the manuscript must specify the scientific name of the jellyfish (Nemopilema nomurai).
2. Page 1, line 10: In the abstract, the scientific name of the jellyfish (Nemopilema nomurai) must be mentioned.
3. Page 2, line 49: It is important to mention in the introduction the scientific name of the jellyfish, and also mention some information about the specific toxicity or the problems caused by Nemopilema nomurai.
4. Page 3, lines 83-86: Usually, hemolytic activity is reported as a logarithmic concentration-response curve. In the figure legend, it should be mention that is hemolytic activity on mice erythrocytes, and if data are shown as mean +/- SD from three independent experiments. Why are the error bars not visible in the graphs?
5. Page 3, line 99: In the first graph of the Figure 3, it must be mention the units of concentration (micrograms/ml?).
6. Page 7, lines 212, 214, 215, 218, 220: It is not necessary to mention the figures in the discussion section.
7. Page 10, lines 371-373: Statistical analysis should be reviewed, because multiple comparison are mentioned, using ANOVA, but a post-hoc test are not mentioned (Tukey or Dunnett?). Student’s test is not a suitable method for these data.
Round 2
Reviewer 2 Report
This manuscript, describing the protective effects of EGCG against the jellyfish Nemopilema nomurai venom, clearly and concisely presents accurate and completed data. It might be accepted in the present form for publication in Toxins journal. The authers made all suggested correction and carefully revised manuscript.
Author Response
Thanks for your positive comments on our manuscript.